# Leucine Supplementation Improves Diastolic Function in HFpEF by HDAC4 Inhibition

**DOI:** 10.3390/cells12212561

**Published:** 2023-11-02

**Authors:** Paula Ketilly Nascimento Alves, Antje Schauer, Antje Augstein, Anita Männel, Peggy Barthel, Dirk Joachim, Janet Friedrich, Maria-Elisa Prieto, Anselmo Sigari Moriscot, Axel Linke, Volker Adams

**Affiliations:** 1Laboratory of Experimental and Molecular Cardiology, TU Dresden, Heart Center Dresden, 01307 Dresden, Germany; paulaketilly@usp.br (P.K.N.A.); antje.schauer@tu-dresden.de (A.S.); antje.augstein@tu-dresden.de (A.A.); anita.maennel@tu-dresden.de (A.M.); peggy.barthel@tu-dresden.de (P.B.); dirk.joachim@tu-dresden.de (D.J.); janet.friedrich@tu-dresden.de (J.F.); axel.linke@tu-dresden.de (A.L.); 2Department of Anatomy, Institute of Biomedical Sciences, University of Sao Paulo, São Paulo 05508000, Brazil; moriscot@usp.br

**Keywords:** ZSF1 rat, HFpEF, leucine, diastolic dysfunction, HDAC4

## Abstract

Heart failure with preserved ejection fraction (HFpEF) is a complex syndrome associated with a high morbidity and mortality rate. Leucine supplementation has been demonstrated to attenuate cardiac dysfunction in animal models of cachexia and heart failure with reduced ejection fraction (HFrEF). So far, no data exist on leucine supplementation on cardiac function in HFpEF. Thus, the current study aimed to investigate the effect of leucine supplementation on myocardial function and key signaling pathways in an established HFpEF rat model. Female ZSF1 rats were randomized into three groups: Control (untreated lean rats), HFpEF (untreated obese rats), and HFpEF_Leu (obese rats receiving standard chow enriched with 3% leucine). Leucine supplementation started at 20 weeks of age after an established HFpEF was confirmed in obese rats. In all animals, cardiac function was assessed by echocardiography at baseline and throughout the experiment. At the age of 32 weeks, hemodynamics were measured invasively, and myocardial tissue was collected for assessment of mitochondrial function and for histological and molecular analyses. Leucine had already improved diastolic function after 4 weeks of treatment. This was accompanied by improved hemodynamics and reduced stiffness, as well as by reduced left ventricular fibrosis and hypertrophy. Cardiac mitochondrial respiratory function was improved by leucine without alteration of the cardiac mitochondrial content. Lastly, leucine supplementation suppressed the expression and nuclear localization of HDAC4 and was associated with Protein kinase A activation. Our data show that leucine supplementation improves diastolic function and decreases remodeling processes in a rat model of HFpEF. Beneficial effects were associated with HDAC4/TGF-β1/Collagenase downregulation and indicate a potential use in the treatment of HFpEF.

## 1. Introduction

Heart failure (HF) is a worldwide burden that affects around 26 million individuals. Approximately half of all HF patients present with a preserved left ventricular ejection fraction (HFpEF; LVEF greater than 50%), and the prevalence is increasing (around 10% per decade) due to an aging population and an increase in other risk factors such as diabetes mellitus, arterial hypertension, and sedentary lifestyle [1,2].

The HFpEF phenotype is characterized by diastolic dysfunction due to impaired left ventricular (LV) relaxation and/or increased LV stiffness; in other words, the heart loses its ability to appropriately store blood during diastole [3]. The condition is accompanied by cellular and structural cardiac alterations such as cardiomyocyte hypertrophy, inflammation, and fibrosis, which ultimately contribute to decreased relaxation of the left ventricle despite the intrinsic contractile capacity of the heart remaining within the normal range [3].

Thus far, most efforts to reduce morbidity and mortality in HFpEF have failed. Only the recently published EMPEROR-trail, which used Empagliflozin [4], a Sodium-Glucose-Transporter 2 inhibitor, and exercise training [5,6], showed beneficial results for HFpEF.

Another promising therapy for HF includes nutritional intervention by leucine supplementation. Amino acids provide approximately 30% of the energy that sustains the contractile function and are used for the maintenance of ionic homeostasis in the heart [7,8]. This may be related to recent findings of leucine supplementation improving mitochondrial fusion, function, and respiration, as well as ATP production during HF [9,10]. Leucine has also been demonstrated to lower arrhythmia [11] and myocardial damage [12]. Interestingly, lower L-leucine levels predict increased cardiovascular death in HFpEF, and the authors proposed that supplementary leucine could be used as an additional treatment in HFpEF [13]. Furthermore, leucine supplementation inhibited doxorubicin-induced ventricular dilatation, increased collagen fiber content, and pathological cardiac remodeling [12].

HDACs are an enzyme family that removes N-acetyl lysine from histone and non-histone proteins. Class II HDACs (HDAC4, HDAC5, HDAC7, and HDAC9) are highly expressed in the heart [14,15]. Recently, it was demonstrated that activation of HDAC4 increased cardiac dysfunction by inducing cardiac hypertrophy and fibrosis [16]. Additionally, the same authors demonstrated that, in myocardial infarction, overexpression of HDAC4 exacerbated cardiac dysfunction and augmented cardiac remodeling and fibrosis [17].

Furthermore, we recently demonstrated for the first time that leucine supplementation significantly reduces HDAC4 expression and positive nuclear localization induced by hindlimb immobilization [18]. These findings imply that HDAC4 inhibition may be involved in leucine’s anti-atrophic effect.

Despite these encouraging results, the mechanisms by which leucine supplementation protects the myocardium remain unclear. Thus, the purpose of this study was to investigate the impact of leucine supplementation on myocardial function in an established HFpEF rat model. Furthermore, we assessed leucine’s effect on important key signaling pathways that control myocardial function, examined mitochondrial respiratory capacity, stress, anabolic, catabolic, and fibrotic markers, as well as HDAC4 protein expression and tissue localization.

## 2. Materials and Methods

### 2.1. Study Design

In total, 34 female ZSF1 rats (Charles River Laboratories, Wilmington, MA, USA) were used in this study, taking into account that HFpEF is predominantly reported in the female phenotype. Obese rats carrying two leptin receptor mutations (fa: facp) develop HFpEF. Single or no mutation leads to a healthy, lean phenotype [19]. The animals were randomized into three groups: Control (*n* = 10, untreated lean rats receiving standard chow throughout the experiment); HFpEF (*n* = 12, untreated obese rats receiving standard chow throughout the experiment), and HFpEF_Leu (*n* = 12, obese rats receiving standard chow enriched with 3% leucine) (Figure 1). All animals were housed under regular conditions including controlled temperature (24 ± 1 °C, 12-h light-dark cycle), and food and water provided *ad libitum*.

Leucine supplementation started at 20 weeks of age, after HFpEF was established in obese animals [19,20,21,22], and the dose was adapted from previous studies [11,12]. Echocardiography was performed at 20, 24, and 32 weeks of age. All animals underwent invasive left ventricular and aortic hemodynamic measurement at the age of 32 weeks prior to organ harvest (Figure 1).

This study was approved by the local animal research council, TU Dresden, and the Landesbehörde Sachsen (TVV 26/2022).

### 2.2. NT-proBNP

Blood serum was separated by centrifugation, and NT-proBNP levels were measured using RatNT-proBNP ELISA (Biomedica Immunoassays, Vienna, Austria) according to the manufacturer’s protocol.

### 2.3. Echocardiography

Rats were anesthetized with isoflurane (1.5–2%) and placed on a controlled warming pad with electrodes that continuously recorded breathing frequency, heart rate, and body temperature. Transthoracic echocardiography was performed using a Vevo 3100 system and a 21-MHz transducer (FUJIFILM VisualSonics Inc., Amsterdam, Netherlands) to assess cardiac function as recently described [19,20,21,22]. For systolic function, B- and M-mode of the parasternal long and short axis were measured. LV wall thickness and cavity diameters were measured in short axis view, both in end-diastole and end-systole, with the M-mode cursor perpendicular to the LV anterior and posterior walls at the level of the papillary muscles. LV structural parameters measured from the short axis view in B-mode were used in the calculation of LVEF. Left ventricle mass was calculated in mg using the formula 1.053 × ((LVID;d + LVPW;d + IVS;d)^3^ − LVID;d^3^) and corrected by the factor 0.8.

Diastolic function was assessed in the apical 4-chamber view using pulse wave Doppler (for measurement of early (E) and atrial (A) waves of the mitral valve velocity) and tissue Doppler (for measurement of myocardial velocity (é and á)) at the level of the basal septal segment in the septal wall of the left ventricle.

Functional parameters (i.e., LV ejection fraction (LVEF) and stroke volume (SV)) and ratios of E/é and E/A) were computed by the Vevo LAB 3.1.1 software.

### 2.4. Invasive Hemodynamics

Prior to organ harvest, invasive hemodynamic measurements were performed as recently described [19]. In anesthetized (i.p. injection of ketamine (105 mg/kg) and xylazine (7 mg/kg)) but spontaneously breathing rats, the right carotid artery was cannulated with a Rat PV catheter (SPR-838, ADInstruments Ltd, Oxford, UK) which was gently placed in the middle of the left ventricle. Pressure-volume loops were recorded under baseline conditions and during transient occlusion of the inferior vena cava by external compression of the vessel to obtain load-independent indexes of contractility and chamber stiffness. The obtained end-systolic and end-diastolic pressure–volume relationships (ESPVR, EDPVR) were fitted to linear and exponential functions, respectively, with the slope Ees indicating contractility and the chamber stiffness constant β displaying the grade of diastolic compliance. To take account of potentially different heart sizes, the left ventricular wall volume (Vw) was used as a normalization factor (β * Vw = βw). Data were recorded and analyzed with LabChart 8 software (ADInstruments Ltd., Oxford, UK).

### 2.5. Left Ventricular Mitochondrial Respiration

The respiratory parameters of the total mitochondrial population were studied in saponin-skinned fibers of myocardial tissue. Respiratory rates were determined by using a Clark electrode (Strathkelvin Instruments, Motherwell, UK) in an oxygraphic cell at 25 °C with continuous stirring. To avoid oxygen diffusion limitation, the oxygen concentration was increased to ~400 µmol/L by adding pure oxygen and was kept above 270 µmol/L throughout the experiment. Left ventricular muscle fibers were isolated in permeabilization solution (SolP) containing, in mmol/L: 2.77 CaK2EGTA, 7.23 K2EGTA, 6.56 MgCl2, 5.7 Na2ATP, 15 phosphocreatin (PCr), 20 taurine, 0.5 DTT, 50 K methane sulfonate, and 20 imidazole (pH 7.1) and incubated for 30 min in SolP with 50 µg/mL saponin. Permeabilized fibers were transferred to respiration solution (SolR) (in mmol/L: 20 taurine, 20 HEPES, 10 KH2PO4, 0.5 EGTA, 3 MgCl2, 0.11 sucrose, and 60 K-lactobionate (pH 7.4)) for 10 min to wash out adenine nucleotides and PCr. All steps were carried out at 4 °C under continuous stirring. Respiration rates of 1–5 mg of skinned fibers were measured at 25 °C in 1 mL of SolR containing 1 mg/mL bovine serum albumin. The following substrates were added sequentially and oxygen consumption was monitored: (I) glutamate (10 mmol/L), malate (2.0 mmol/L), (complex I state 2 respiration); (II) adenosine diphosphate (5.0 mmol/L; measure of complex I oxidative phosphorylation); (III) octanoylcarnitine (0.2 mmol/L; measure complex I activated by fatty acid oxidation); (IV) cytochrome C (10 µmol/L; test for membrane integrity); (V) succinate (10 mmol/L; oxidative phosphorylation of complex I + II); (VI) rotenone (0.5 mmol/L; oxidative phosphorylation of complex II); (VII) FCCP (0.5 µmol/L, maximal uncoupled complex II respiration); (VIII) antimycin A (2.5 µmol/L, complex III inhibitor). After the experiment, fiber bundles were blotted dry and weighed. Rates of respiration are given in nanomoles O2 per second per mg wet weight.

### 2.6. Enzymes Activity

Left ventricular tissue was homogenized in Relax buffer (90 mmol/L HEPES, 126 mmol/L potassium chloride, 36 mmol/L sodium chloride, 1 mmol/L magnesium chloride, 50 mmol/L EGTA, 8 mmol/L ATP, 10 mmol/L phosphocreatin, pH 7.4) and aliquots were used for enzyme activity measurements. Enzyme activities for citrate synthase (EC 2.3.3.1), pyruvate kinase (EC 2.7.1.40), and lactate dehydrogenase (EC 1.1.1.27) were measured spectrophotometrically as recently described [23,24,25,26,27].

### 2.7. Western Blotting

Snap-frozen left ventricular tissue was homogenized in Relax buffer containing a protease inhibitor mix (Inhibitor mix M, Serva, Heidelberg, Germany) and sonicated. Protein concentration was determined (BCA assay, Pierce, Bonn, Germany), and aliquots (10–40 µg) were separated by SDS-polyacrylamide gel electrophoresis.

Proteins were transferred to a polyvinylidene difluoride membrane. To verify a homogeneous loading, membranes were stained with Ponceau S. Next, membranes were blocked with 5% non-fat dry milk in Tris-buffered saline with Tween (TBS-T, 0.5 M NaCl, 50 mM Tris-HCl pH 7.4, 0.1% Tween 20) for 1 h at room temperature, followed by overnight incubation at 4 °C with primary antibody. Primary antibodies were as follows: rabbit anti-MAFbx (1:1000; Abcam, #ab168372), mouse anti-total OXPHOS (1:250; Abcam, # ab110413), rabbit anti-AKT (1:1000; Proteintech, #10176-2-AP), rabbit anti-AMPKα (1:1000; Cell Signaling, #2532), rabbit anti-pAMPKα (1:1000; Cell Signaling, #2531), rabbit anti-PKA (1:1000; Cell Signaling, #4782), rabbit anti-pPKA (1:1000; Sigma, #SAB4503969), mouse anti-MuRF1 (1:250; Santa Cruz, #sc-398608), mouse anti-myogenin (1:250; Santa Cruz, #sc-12732), mouse anti-eIF2Bδ (1:250; Santa Cruz, #sc 9981), rabbit anti-myostatin (1:1000; Proteintech, #19142-1-AP), rabbit anti-HDAC4 (1:1000; Cell Signaling, #7628) and rabbit anti-GAPDH (1/30,000; HyTest Ltd., Turku, Finland). After washing (3 times for 5 min, TBS-T), the membranes were incubated with a horseradish peroxidase-conjugated secondary antibody, and specific bands were visualized using enzymatic chemiluminescence (Super Signal West Pico, Thermo Fisher Scientific Inc., Bonn, Germany). Densitometry analyses were performed using a 1D scan software package (Bio-1D version 15.08b, Vilber Lourmat, Eberhardzell, Germany), and the GAPDH signal was used to normalize variations in loading.

### 2.8. RNA Extraction and Quantitative Real-Time PCR

Total RNA was isolated from LV tissue using Qiazol reagent and the miRNeasy Mini Kit (Qiagen, Hilden, Germany) following the standard protocols. cDNA was synthesized with the Revert AID™ H Minus First Strand Synthesis Kit (Thermo Scientific, Braunschweig, Germany) using oligo-dT primers. Real-time PCR was performed using the CFX384TM Real-Time PCR System (Bio-Rad Laboratories GmbH, Feldkirchen, Germany) and Maxima SYBR Green qPCR Kit (Thermo Scientific, Braunschweig, Germany). The PCR program for all primer sets was as follows: 95 °C for 8 min prior to 40 amplification cycles, each consisting of 95 °C for 10 s, 58 °C for 15 s, and 72 °C for 30 s, with a final extension step at 72 °C for 2 min. Melting point analysis was performed to prove the identity of the PCR products. Relative quantification of gene expression was calculated using the ∆∆CT method with Polr2a and Rpl-32 as housekeeping genes using BioRad CFX Maestro 1.1 version 4.1.2433.1219 (Bio-Rad Laboratories GmbH, Feldkirchen, Germany). Specific primer sequences are listed in Appendix A.

### 2.9. Immunohistochemistry

Heart cryo-sections (10 µm) were fixed at room temperature in 4% PFA (Paraformaldehyde) for 10 min, washed (3 times for 5 min) with phosphate-buffered saline (PBS), and then permeabilized with PBS-T (PBS with 0.1% Triton X-100) for 30 min. Subsequently, the sections were incubated with blocking solution (DAKO, #X0909) for 1 h at room temperature and incubated at 4 °C with primary antibody rabbit anti-HDAC4 (1:100; Cell Signaling, #7628) in diluent solution (DAKO, #S3022) overnight. Thereafter, slides were washed with PBS (3 times for 5 min), incubated with the secondary antibody (1:500 Alexa 568 Donkey Anti-Rabbit, Invitrogen, #A10042) in diluent solution for 1 h and subsequently washed (3 times for 5 min) in PBS. The slides were incubated with DAPI (4′,6-diamidino-2-phenylindole) for 5 min and mounted with coverslips using Mount Fluor (Protaqs, Potsdam, Germany). Digital acquisitions and the nuclear colocalization analysis were performed using microscopy equipment (Echo RVL-100-M Revolve Fluorescence Microscopy). For cross-sectional area measurements (CSA), the myocardial heart sections were stained with Collagen IV (#AB756P, Millipore, Darmstadt, Germany), and ImageJ (v. 1.45s, National Institutes of Health, Bethesda, MD, USA) was used to measure the heart fibers. Approximately 1000 fibers per group were analyzed.

For fibrosis measurements, paraffin-embedded heart sections (3 µm) were stained with picrosirius red, and perivascular fibrosis around arteries, expressed as perivascular fibrosis ratio (PFR), was quantified as described by Dai and colleagues [28]. PFR was calculated by dividing the area of perivascular fibrosis by the area of the vascular wall and averaging it over all measurable pictures of arteries acquired from a segment.

### 2.10. Data Analysis

Data analyses were performed using Prism Software (GraphPad Prism 7.0). Multiple comparisons were performed using either one-way ANOVA followed by Tukey’s posthoc test (for parametric data) or the Kruskal–Wallis test of one-way ANOVA followed by Dunn’s post hoc test (for non-parametric data). Data are presented as mean ± SEM, and *p* < 0.05 was considered significant.

## 3. Results

### 3.1. Impact of Leucine Supplementation on Biometric Features, Myocardial Function and Hemodynamics

Table 1 shows the overall data of biometric features, echocardiography, and hemodynamic measurements. At 32 weeks, both HFpEF groups presented higher body, heart, and kidney weight compared to the lean control. The lung wet weight was significantly higher (a sign of congestion) in HFpEF but not HFpEF_Leu, as were plasma levels of NT-proBNP. The lactate serum levels were increased in both HFpEF groups compared to the control (Table 1).

Indicators of myocardial hypertrophy LV mass and septum thickness were decreased in HFpEF_Leu compared to HFpEF (Table 1). Left ventricular anterior and posterior wall thickness, as well as the inner diameter, were increased in both HFpEF groups compared to the control group (Table 1).

Systolic function was preserved in all groups at all times, as indicated by the similar ejection fraction (Table 1). Systolic blood pressure (LVESP) was increased in both HFpEF groups, but diastolic blood pressure (LVEDP), which was enhanced in HFpEF, was normalized by leucine supplementation.

Diastolic function was significantly improved after four weeks of leucine supplementation (E/é: 11% reduction vs. HFpEF) and normalized after 12 weeks of treatment (E/é: 20% reduction vs. HFpEF) (Table 1 and Figure 2). LV stiffness was significantly decreased in HFpEF_Leu compared to HFpEF (Table 1).

### 3.2. Impact of Leucine Supplementation on Myocardial Fibrosis and Fiber Size

Given that the hemodynamics assay revealed a decrease in LV stiffness in HFpEF_Leu compared to HFpEF, we proceeded to histology analysis. Perivascular fibrosis was comparable between all groups, as shown in Figure 3B.

In addition, we assessed the LV fiber cross-sectional area and found a significant increase in HFpEF when compared to the control, which was reduced by leucine supplementation (Figure 3C), confirming the decrease in septum thickness observed in the hemodynamics assay. These findings are supported by fiber CSA distribution, which shows a high frequency of large fibers in untreated HFpEF, while HFpEF_Leu fiber size is in line with the control (Figure 3D).

### 3.3. Impact of Leucine Supplementation on Myocardial Stress and Fibrosis

Compared to the control, the expression of ANP mRNA was upregulated in HFpEF and tendentially reduced in HFpEF_Leu (*p* = 0.098, Figure 4A). BNP mRNA levels were upregulated in both HFpEF groups (Figure 4B). The expressions of Nox2 were comparable between all groups (Figure 4C).

Transforming Growth Factor Beta 1 (TGF-β1) expression levels were tendentially increased in HFpEF (*p* = 0.077) but back to the control in the leucine-treated group (Figure 4D). Expression of Collagen Type I a1 (Col1a1, Figure 4E) and Collagen Type III a1 mRNA (Col3a1, Figure 4F) were increased in HFpEF and significantly reduced in HFpEF_Leu.

Accordingly, the expression of lysyl oxidase (LOX), a key mediator of collagen maturation, was tendentially decreased in HFpEF_Leu compared to untreated HFpEF (*p* = 0.061) (Figure 4G).

### 3.4. Impact of Leucine Supplementation on Mitochondrial Respiratory Function

The sequential addition of complex I and II substrates/inhibitors enabled complex- and substrate-specific monitoring of oxygen consumption (Figure 5).

Basal respiration (only substrate and no ADP added, state 2) did not differ between all three groups (Control: 0.066 ± 0.004 nmol/s/mg, HFpEF: 0.057 ± 0.004 nmol/s/mg and HFpEF_Leu: 0.051 ± 0.004 nmol/s/mg, n.s., Figure 5A). Measurement of complex I state 3 (glutamate/malate) showed a significantly decreased respiration rate in HFpEF compared to the control (Control: 0.163 ± 0.009 nmol/s/mg, HFpEF: 0.123 ± 0.003 nmol/s/mg and HFpEF_Leu: 0.154 ± 0.017 nmol/s/mg, ** *p* < 0.005 vs. Control, Figure 5B); likewise, the same effect was observed following the addition of octanoyl-carnitine (Control: 0.165 ± 0.011 nmol/s/mg, HFpEF: 0.127 ± 0.006 nmol/s/mg and HFpEF_Leu: 0.156 ± 0.019 nmol/s/mg, * *p* < 0.05 vs. Control, Figure 5C).

Subsequent addition of succinate and rotenone (state 3 of complex II, Figure 5D) resulted in decreased oxygen consumption in HFpEF compared to the control. Leucine normalized oxygen consumption (Control: 0.132 ± 0.007 nmol/s/mg, ** HFpEF: 0.101 ± 0.004 nmol/s/mg vs. Control and * HFpEF_Leu: 0.126 ± 0.009 nmol/s/mg vs. HFpEF).

Respiratory control ratio (RCR) was increased in HFpEF_Leu compared to HFpEF (Control: 2.29 ± 0.12 nmol/s/mg, HFpEF: 2.08 ± 0.13 nmol/s/mg and HFpEF_Leu: 2.94 ± 0.21 nmol/s/mg, * *p* < 0.05 vs. HFpEF, Figure 5E). These findings suggest improved cardiac mitochondrial respiratory performance in HFpEF after leucine supplementation.

### 3.5. Impact of Leucine Supplementation on Protein Expression of Mitochondrial Complexes

In order to analyze whether the improved mitochondrial respiratory function in HFpEF_Leu was due to an altered mitochondrial complex amount, we analyzed the protein expression of complex I-V (Figure 6).

As shown in Figure 6, there were no significant differences in protein expression of mitochondrial complexes between all groups.

### 3.6. Impact of Leucine Supplementation on Myocardial Metabolism

Specific marker enzymes were analyzed in left ventricular tissue to gain insight into whether leucine supplementation changed metabolic pathways (Table 2).

The activities of citrate synthase were comparable between all groups. Lactate dehydrogenase was decreased in HFpEF compared to control. Pyruvate kinase was decreased in both HFpEF groups compared to the control.

### 3.7. Impact of Leucine Supplementation on Catabolic and Anabolic Markers

It has been described that leucine supplementation promotes protein synthesis in the skeletal muscle by increasing mTOR signaling while simultaneously inhibiting protein degradation (by E3 ligases inhibition/MAFbx and MuRF1) [18,29,30]. Therefore, the main catabolic and anabolic signaling indicators were measured in the left ventricle of all three groups (Figure 7).

Regarding the catabolic pathway, myogenin, a transcription factor of MAFbx and MuRF1, was downregulated in both HFpEF groups (Figure 7A). The expression of myostatin, an alternative upstream inducer of MAFbx and MuRF1, showed no differences between the groups (Figure 7B). In addition, neither MAFbx nor MuRF1 expression was altered between the groups (Figure 7C and D, respectively).

Concerning the anabolic pathway, we analyzed the expression of one upstream marker (AKT, Figure 7E) and one downstream marker (eIF2Bδ, Figure 7F) of mTOR. AKT and eIF2Bδ expression were decreased in HFpEF compared to the control and even lower in HFpEF_Leu compared to both the control and HFpEF groups.

### 3.8. Impact of Leucine Supplementation on HDAC4 Modulation

Looking for the underlying mechanisms of the beneficial effect on diastolic function mediated by leucine supplementation, HDAC4, a protein highly expressed in the heart and involved in cardiac dysfunction [16,17,31], was investigated.

HDAC4 is inactivated by phosphorylation, and two important kinases mediating the inactivation of HDAC4 are AMPKα and PKA (Figure 8A and B, respectively). While overall PKA expression was comparable between all groups, we observed phosphorylated PKA upregulation in HFpEF_Leu (Figure 8B). HDAC4 protein expression was upregulated in HFpEF and normalized in HFpEF_Leu (Figure 8C). These results suggest that HDAC4 is responsive to leucine, and PKA-HDAC4 signaling might mediate improved diastolic function.

HDAC4 inactivation is accompanied by nuclear exportation, which reduces its action on target genes. Histological analysis of HDAC4 distribution in left ventricular tissue revealed a significantly increased number of HDAC4-positive nuclei in HFpEF that were reduced by leucine supplementation (Figure 9B), strengthening the impression that HDAC4 might be involved in the observed beneficial effects.

## 4. Discussion

The present study demonstrates that leucine supplementation improves diastolic function in a rat model of HFpEF. This was accompanied by reduced left ventricular stiffness, fibrosis, and hypertrophy (Figure 10). As potentially underlying mechanisms of the observed beneficial effects, leucine improved cardiac mitochondrial respiratory function and suppressed cardiac HDAC4 activation (Figure 10).

Treatment of HFpEF remains challenging. Apart from SGLT2i [4] and exercise training [5,6], most efforts to improve morbidity and mortality in HFpEF have failed so far (for review, see [32]). In this context, we propose an important approach for leucine supplementation in order to analyze whether a nutritional strategy might be beneficial for treating HFpEF.

It is important to highlight that leucine supplementation improved diastolic function and was accompanied by reduced cardiac remodeling, indicating that an established HFpEF responds to the supplementary treatment. Since most patients are diagnosed after the disease has already developed, these findings encourage the use of leucine as a potential therapeutic treatment for HFpEF.

In the literature, several benefits of leucine supplementation in HF conditions are documented, including atherosclerosis development prevention, decreased damage induced by myocardial ischemia/reperfusion (I/R) injury, and inhibited doxorubicin-induced pathological cardiac remodeling [9,10,11,12,13]. However, underlying molecular mechanisms are not fully understood.

Tissue remodeling and the presence of fibrosis disrupt the architecture of the myocardium and increase the development of cardiac dysfunction, affecting patients’ clinical outcomes [33,34,35]. Thus, addressing fibrotic pathways might help to improve the therapy of heart failure patients. Impaired passive myocardial relaxation in HFpEF has mainly been attributed to concentric remodeling and fibrosis [20]. Previously, Fidale et al. reported a significant impact of leucine supplementation on cardiac remodeling in heart failure induced by doxorubicin [12]. While extracellular matrix remodeling was increased in the doxorubicin group, supplementation with leucine prevented it. In line with these findings, we found that leucine supplementation had significantly beneficial effects on LV remodeling. Improved diastolic function was accompanied by decreased left ventricular stiffness, which might be related to the downregulation of main markers involved in cardiac fibrosis (Col1a1, Col3a1, and TGF-β1). Since we found no differences in perivascular fibrosis, we speculate that the remodeling impact of leucine supplementation and the increased fibrosis revealed by PCR (Figure 4) may be restricted to the interstitial cardiac tissue, as represented in the interstitial pictures (Figure 3).

The mechanism underlying leucine’s effect on fibrosis and hypertrophy might likewise be linked to HDAC4 suppression [16]. TGF-β binds and thereby activates the TGF-β receptor, which phosphorylates Smad2 and Smad3. These interact with Smad4, which translocates into the nucleus to induce the expression of fibrotic targeted genes [36,37], including collagen 1 and 3 [38,39,40]. The essential role of Smad3 in fibrotic remodeling has been confirmed by a study using Smad3-null mice. Bujak et al. found that mice with Smad3 deficiency had reduced fibrotic remodeling after myocardial infarction [41]. HDAC4 has been shown to be involved in several TGF-β/Smad pathway activities in a variety of cells, including osteoblasts [42], fibroblasts [43], and skeletal muscle [44]. In fact, HDAC4 is required for TGF-β1-induced myofibroblastic differentiation [43]. Interestingly, TGF-β regulates HDAC4 via TGF-β1- Smad3 signaling [42,45]. Thus, the TGF-β1/Smad3 axis and HDAC4 might act together in the fibrotic molecular mechanism, which might also be related to leucine’s antifibrotic effect observed in this study.

Morio and collaborators [10] recently demonstrated that leucine increases mitochondrial fusion, size, and volume in the cardiac tissue of high-fat diet-induced obese animals submitted to ischemia/reperfusion injury. However, the authors did not investigate molecular mechanisms of leucine-induced mitochondrial fusion. In the present study, leucine supplementation improved cardiac mitochondrial respiration, as the oxygen consumption of complex II was restored by leucine supplementation. Regarding complex I, the wide variety of respiratory capacity in the leucine-treated group did not lead to statistical differences compared to untreated HFpEF and might reflect the heterogeneous response to leucine. This result was not accompanied by alterations in the overall cardiac mitochondrial respiratory complex amount. However, cardiac levels of HDAC4 were significantly downregulated by leucine supplementation. HDAC4 has been described as a key protein promoting myocardial damage [16,17,46,47,48], and HDAC inhibition has demonstrated protective effects on mitochondrial homeostasis and performance [47]. Thus, enhanced mitochondrial respiratory function might have been a result of HDAC4 inhibition. Although not addressed in the present study, leucine supplementation has been demonstrated to enhance the mitochondrial energetic status of macrophages by boosting ATP production, which might be another manner in which leucine improves mitochondrial cardiac performance [9].

We also measured specific enzyme activities that are modulated by HFpEF. CS is a marker for mitochondria content, and no significant differences were observed between groups, correlating with no changes in mitochondrial respiratory complex protein expression as determined by Western blot (Figure 6). Concerning LDH activity, we observed a decrease in HFpEF in untreated animals, which might be attributed to negative feedback by an increase in blood serum lactate levels (Table 1). In fact, LDH is responsible for the production of lactate, and several studies have demonstrated its role in maintaining the metabolic energy balance in HFpEF [49]. Finally, we measured PK activity, which catalyzes the last step of glycolysis by converting phosphoenolpyruvate into pyruvate, and it is an important regulator of glycolytic flux in a failing heart. Both HFpEF groups showed lower levels of PK, indicating a metabolic imbalance in the model, which might be associated with the accumulation of pyruvate and other glycolytic intermediates commonly observed in myocardial from patients with heart failure [50].

The main mechanism of action postulated for leucine is a stimulation of the mammalian target of rapamycin (mTOR), which is also known as a potent activator of muscle protein synthesis [51]. In HF patients, a metabolic imbalance between the rates of protein synthesis (anabolic pathway) and protein degradation (catabolic pathway) has been described, with the latter prevailing over anabolic hormones [52]. In fact, when we looked at cardiac protein levels of key players of the mTOR pathway (AKT and eIF2Bδ), we found a downregulation in HFpEF, which was not altered by leucine supplementation. This suggests that the pro-trophic mTOR activation pathway by leucine is not operating in this model.

Myogenin protein levels were reduced in both HFpEF groups compared to the control. The mechanisms involved in myogenin reduction during heart failure are not completely understood. However, our findings support previous work performed in cell culture, where doxorubicin-induced cardiotoxicity was associated with suppressed myogenin expression [53]. Although myogenin plays an important role in muscle atrophy, as a transcription factor of MAFbx and MuRF1, it also acts as a myogenic regulatory factor, assisting the differentiation of satellite cells and myotubes formation [54,55]. In response to a muscle injury and inflammation, myogenic differentiation is part of the muscle regenerative process and is negatively affected by myogenin reduction, contributing to the progression of muscle damage [54,55,56]. In this regard, the current study’s findings of decreased myogenin protein levels in both HFpEF groups might be associated with the HFpEF muscle inflammatory burden.

Since HFpEF is a highly clinically relevant disease, it is important to point out how leucine supplementation has been evaluated as a translational treatment. Leucine supplementation has long been used as a therapeutic option, mostly for skeletal muscle [57,58,59,60]. In discussions about the optimal dose response, difficulties in measuring the intended outcomes and timing of leucine supplementation are among the barriers that have slowed down its application in the clinical setting [57,58,59,60,61]. On the other hand, leucine supplementation can be a feasible strategy for treating patients with HFpEF because it is regularly consumed by individuals to increase muscle mass and improve exercise performance. It is easily delivered, and no significant adverse effects have been documented when taken at recommended doses. The International Society of Sports Nutrition recommends that a healthy adult ingest 1–3 g of leucine evenly distributed every 3–4 h across the day [62]. However, a recent study with older adults indicates that elderly persons may require more than double the current levels [63].

To put a comparison involving rodents and humans into context, high dosages of leucine (2.5 g/kg) have traditionally been employed in rodent studies, either by gavage or by adding to the food [11,12,64,65]. The dose employed in this study is around 0.7 g/animal/day, which is equivalent to around 100 g/person/day for a 70 kg human. We recognize that more study is required to better relate rodent and human contexts, starting with lowering the leucine curve dose. In fact, studies have demonstrated that low dosages of leucine may lead to increases in muscle protein synthesis (MPS) comparable to high doses [66,67]. In older women, for example, 1.5 g of LEU-enriched EAA (essential amino acids) (0.6 g LEU) produced equivalent increases in MPS as 6 g LEU-enriched EAA (2.4 g LEU) [66]. Ultimately, as mentioned above, evidence on the leucine optimal dose is unclear and remains a research challenge, particularly in HF and older patients.

Takata et al. proposed a protocol for testing the efficacy and safety of a BCAA (branched-chain amino acids) preparation used in combination with cardiac rehabilitation for patients with chronic heart failure [68]. The authors plan to give the patients a BCAA cocktail containing 1144 mg of L-valine, 1904 mg of L-leucine, and 952 mg of L-isoleucine, twice a day and to analyze a series of parameters, including peak VO2, left ventricular EF in the echocardiogram, muscle strength and absolute values of interleukin-6 (IL-6), and tumor necrosis factor (TNF-α) [68]. This investigation will be useful in getting a better understanding of BCAA dosage in humans. However, another study had already supplemented humans with leucine (5 g/d for three weeks) and discovered that leucine inhibited macrophage foam-cell formation through mechanisms related to the metabolism of cholesterol, triglycerides, and energy production [69]. Although recent promising studies associating BCAA and HF disease are emerging [70,71], further research addressing the leucine dose-curve and mechanism will be necessary to gain a deeper understanding.

### Limitations

Several limitations of this study have to be mentioned. First, the limited number of experimental animals led to a high distribution regarding several parameters. This partly resulted in statistical insignificance or only tendencies, although a direction was apparent. Second, we have not addressed all of the components involved in the various pathways discussed, such as Smad3 in the panel of fibrotic markers. Further analyses (for example, using cell culture models on ventricular cardiomyocytes and fibroblasts) will be necessary to deepen our understanding of underlying pathways regarding the effects of leucine.

## 5. Conclusions

Our data demonstrate that leucine improves diastolic function in HFpEF. This was accompanied by reduced remodeling and involved HDAC4/TGF-β1/Collagenases downregulation.

## Figures and Tables

**Figure 1 cells-12-02561-f001:**
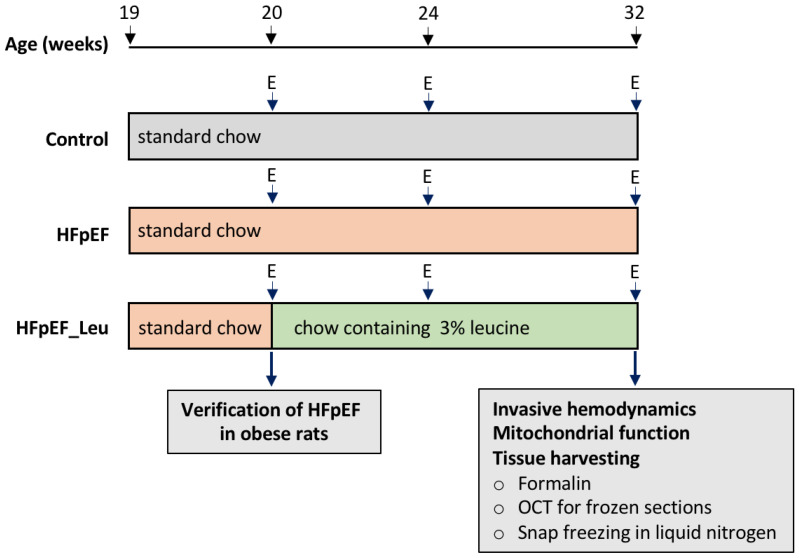
Study design. A total number of 34 ZSF1 female rats was included. At the age of 20 weeks, echocardiography was performed to confirm the development of HFpEF in the ZSF1 obese animals. Thereafter, the ZSF1-obese animals were randomized either into a placebo group (HFpEF *n* = 12, standard chow) or a treatment group (HFpEF_Leu *n* = 12, standard chow enriched with 3% leucine). ZSF1-lean animals served as a healthy control group receiving standard chow (Control, *n* = 10). Follow-up and final echocardiography were performed at the age of 24 and 32 weeks, respectively. Left ventricular hemodynamics were measured prior to organ harvest. The myocardium and other organs were collected for histological and molecular analyses. E = echocardiography.

**Figure 2 cells-12-02561-f002:**
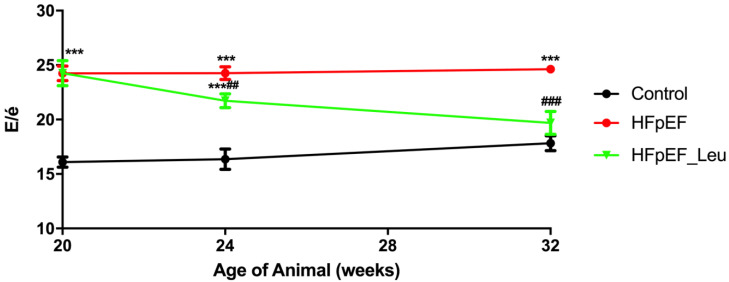
Time course of diastolic function showing a decreased E/é ratio after 4 and 12 weeks of leucine supplementation. *** *p* < 0.001 vs. Control and ## *p* < 0.005 and ### *p* < 0.001 vs. HFpEF.

**Figure 3 cells-12-02561-f003:**
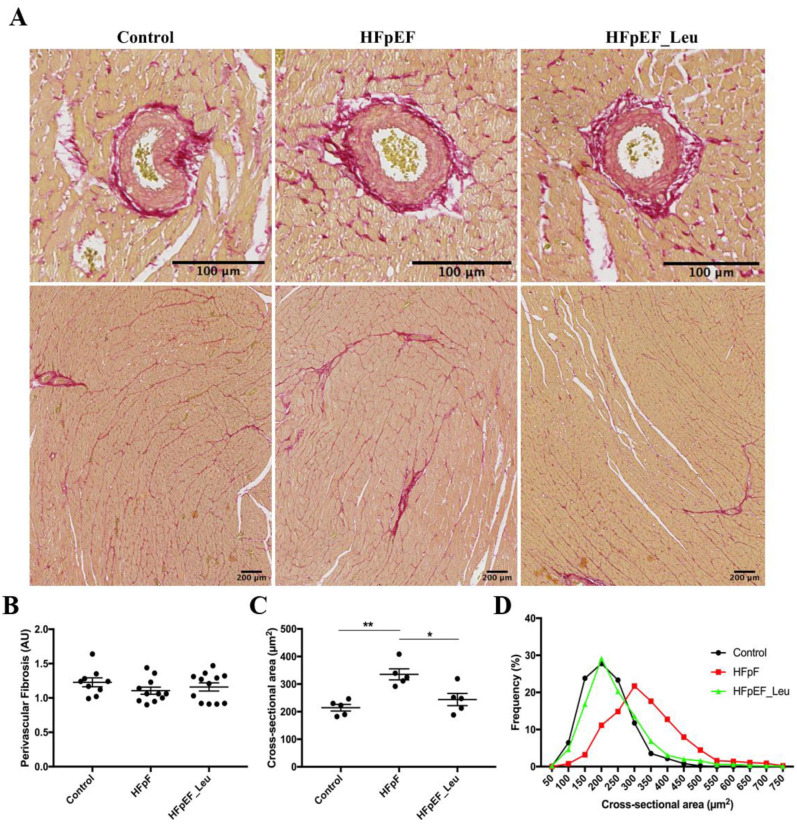
Impact of leucine supplementation on myocardial fibrosis and fiber size. Representative picrosirius staining photomicrographs in LV (**A**). Perivascular fibrosis was comparable between all groups (**B**). Fiber cross-sectional areas were significantly increased in HFpEF and reduced in HFpEF_Leu (**C**). The fiber CSA distribution of control, HFpEF, and HFpEF_Leu indicated a high frequency of large fibers in HFpEF, while HFpEF_Leu fiber size is comparable to the control (**D**). * *p* < 0.05 and ** *p* < 0.005.

**Figure 4 cells-12-02561-f004:**
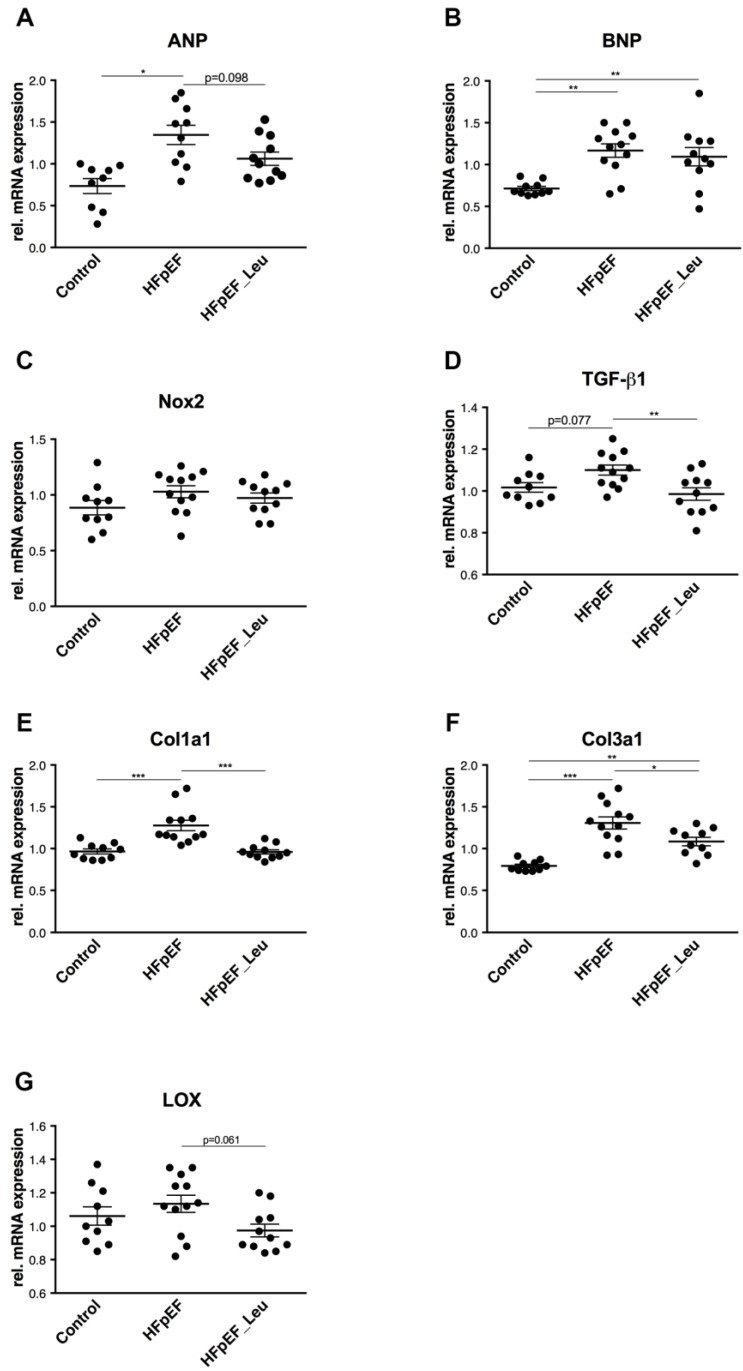
Impact of leucine supplementation on left ventricular stress and fibrosis. mRNA levels of atrial natriuretic peptide (ANP), (**A**) were significantly increased in HFpEF. Expression of brain natriuretic peptide (BNP), (**B**) was increased in both HFpEF groups compared to the control. mRNA levels of NOX2 (**C**) were comparable between all groups. TGF-β1 (**D**) expression was reduced in HFpEF_Leu compared to HFpEF. Collagenase 1 (Col1a1), (**E**) and Collagenase 3 (Col3a1), (**F**) expression levels were increased in HFpEF and decreased in HFpEF_Leu. LOX expression (**G**) was not significantly altered between all groups. * *p* < 0.05, ** *p* < 0.005 and *** *p* < 0.001.

**Figure 5 cells-12-02561-f005:**
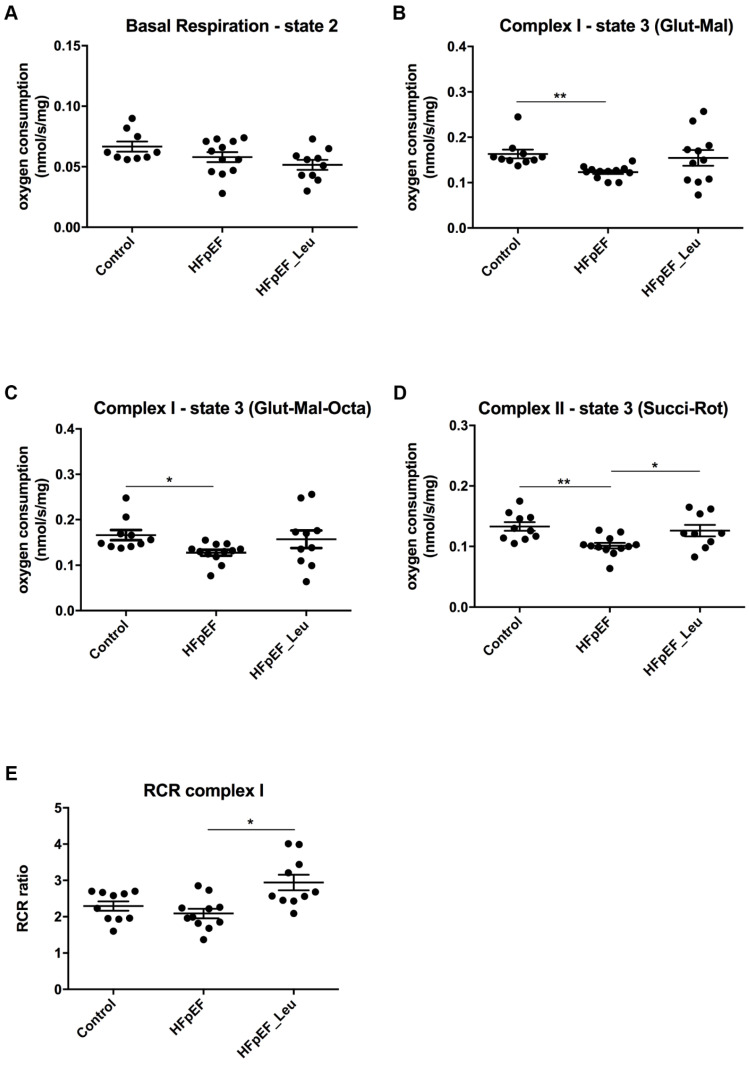
Effect of leucine supplementation on mitochondrial respiratory function. Basal respiratory state 2 (**A**) was similar between all groups. Stimulation of complex I with glutamate/malate (Glut/Mal), (**B**) and with octanoyl-carnitine (Glut-Mal-Octa), (**C**) resulted in a significant decrease in oxygen consumption in HFpEF. Following, stimulation of complex II with succinate (**D**) revealed a significant decrease in oxygen consumption in HFpEF, while leucine treatment normalized it. The respiratory control ratio was increased in HFpEF_Leu compared to HFpEF (**E**). * *p* < 0.05 and ** *p* < 0.005.

**Figure 6 cells-12-02561-f006:**
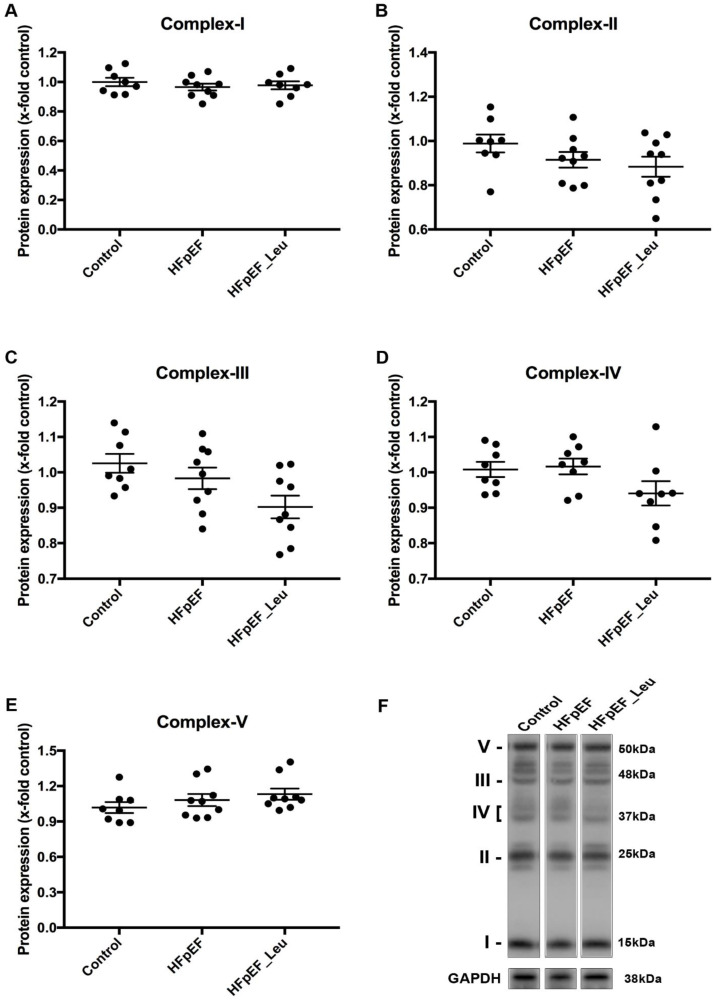
Effect of leucine supplementation on cardiac mitochondrial complex protein expression of HFpEF animals. Protein levels of complex I (**A**), II (**B**), III (**C**), IV (**D**), and V (**E**) were unaltered between all groups. A representative Western blot is depicted (**F**).

**Figure 7 cells-12-02561-f007:**
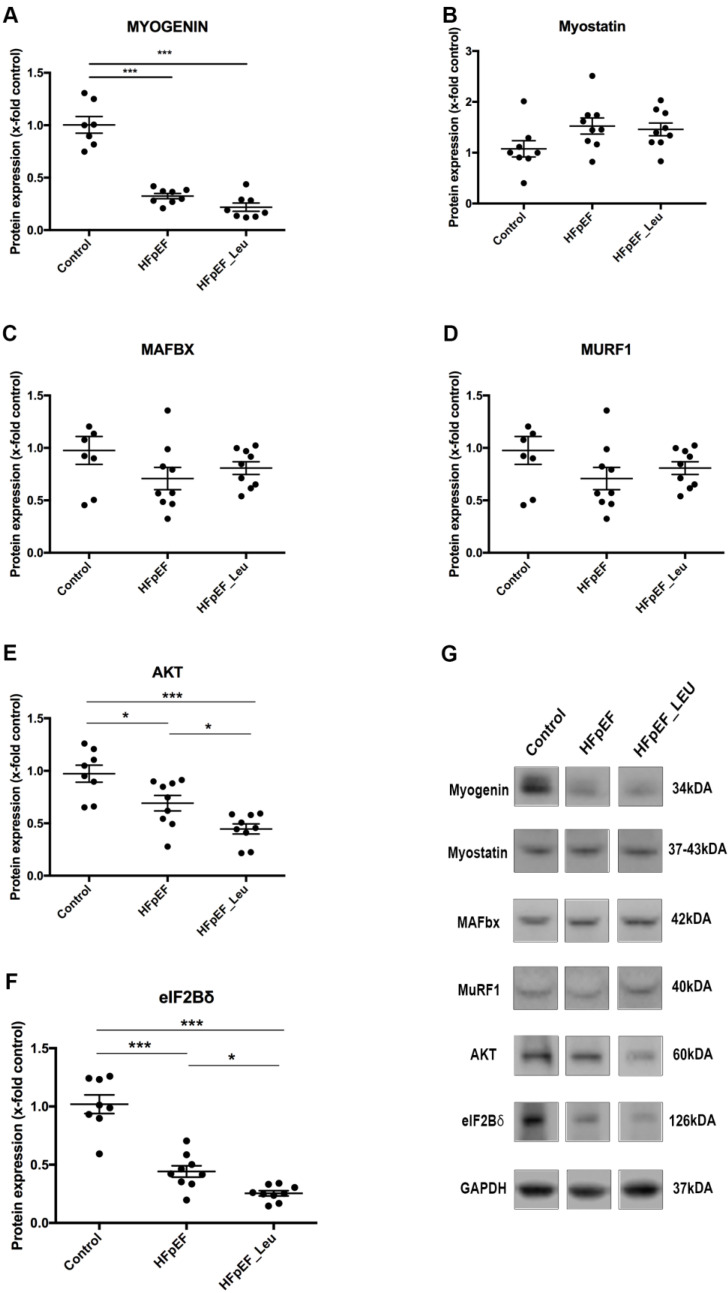
Impact of leucine supplementation on the left ventricular protein turnover. Protein levels of myogenin (**A**) were decreased in both HFpEF and HFpEF_Leu. Myostatin (**B**), MAFbx (**C**), and MuRF1 (**D**) were unaltered between all groups. Protein levels of AKT (**E**) and eIF2Bδ (**F**) were decreased in both HFpEF and HFpEF_Leu. Representative Western blots are depicted (**G**). * *p* < 0.05 and *** *p* < 0.001.

**Figure 8 cells-12-02561-f008:**
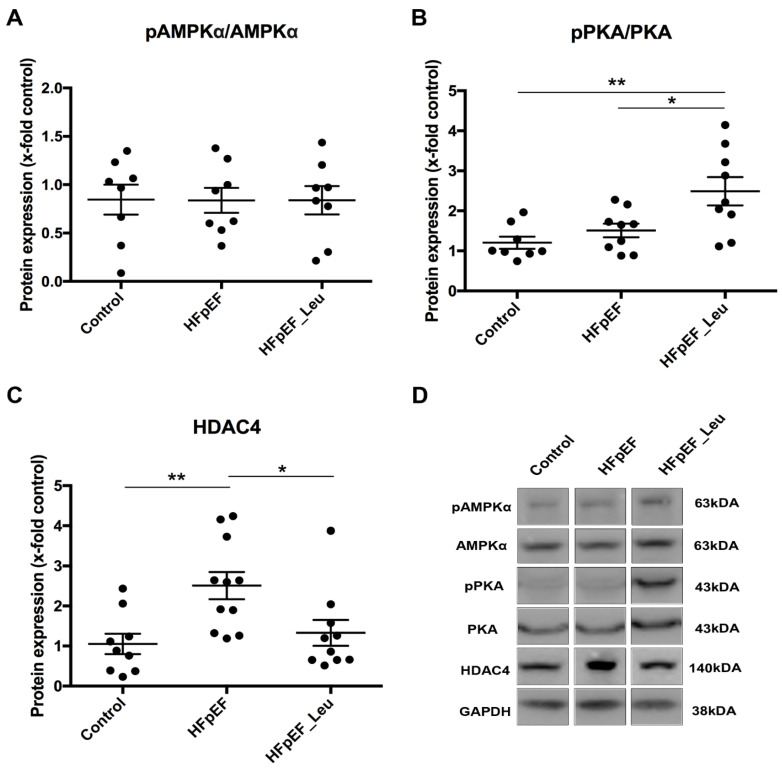
Effect of leucine supplementation on HDAC4 expression and upstream kinases. Protein levels of 5’ AMP-activated protein kinase alfa (AMPKα), (**A**) were unaltered between groups. Protein kinase A (PKA) (**B**) protein levels were increased in HFpEF_Leu, while histone deacetylase 4 (HDAC4) (**C**) was decreased in the same group. Representative bands are depicted (**D**). * *p* < 0.05 and ** *p* < 0.005.

**Figure 9 cells-12-02561-f009:**
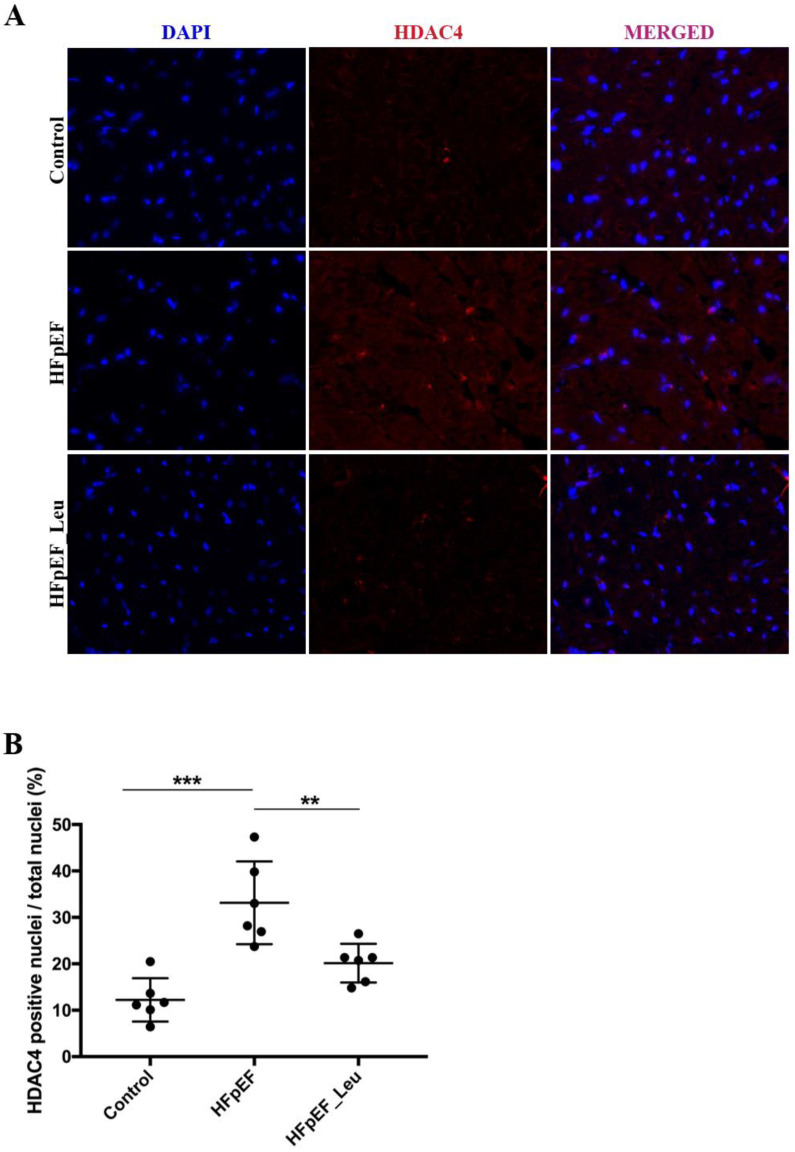
Effect of leucine supplementation on HDAC4 positive nuclei. (**A**) Representative immunofluorescence photomicrographs of HDAC4 in left ventricular tissue, HDAC4 (red), DAPI (blue, used for nuclei identification), and colocalization (merged, pink). (**B**) Percentage of HDAC4 positive nuclei per total nuclei (approximately 300 nuclei per animal were counted) (*n* = 6 per group). ** *p* < 0.005 and *** *p* < 0.001.

**Figure 10 cells-12-02561-f010:**
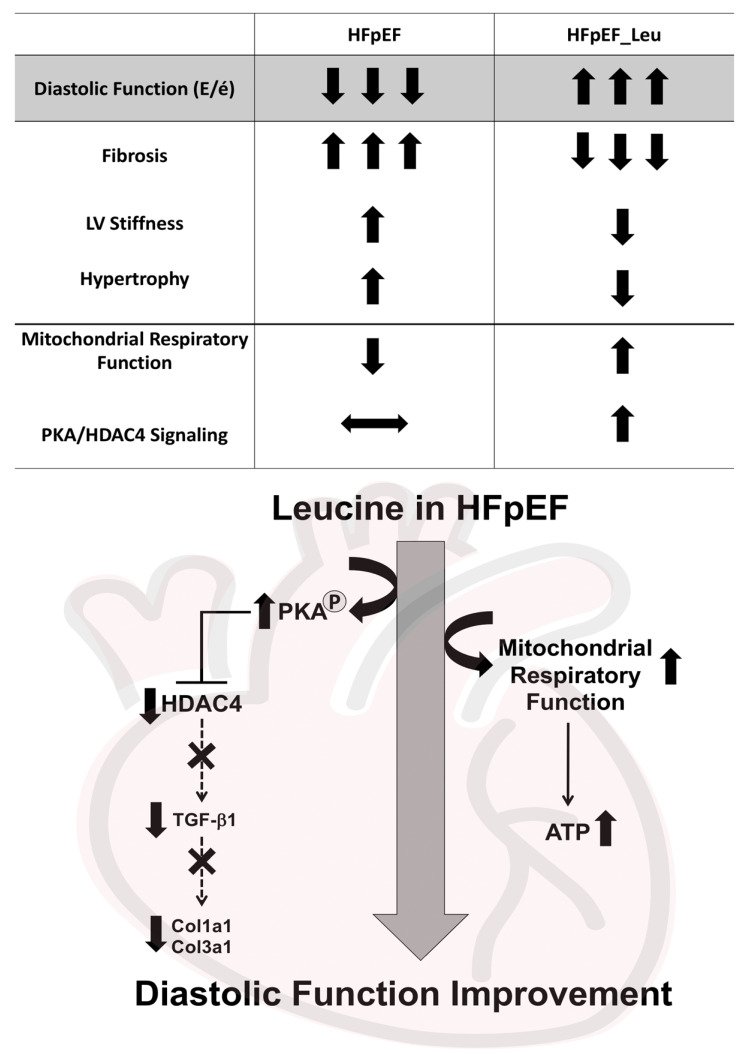
A summary of the beneficial impacts of leucine supplementation on the pathophysiology of HFpEF—mode of action and possible underlying mechanisms in the myocardium.

**Table 1 cells-12-02561-t001:** ZSF1 rat features at 32 weeks of age.

Biometric Feature			
Parameter	Control (*n* = 10)	HFpEF (*n* = 12)	HFpEF_Leu (*n* = 12)
Body Weight (g)	259.6 ± 2.81	522.5 ± 11.85 ***	485.8 ± 5.47 ***^##^
Tibia Length (TL, mm)	35.31 ± 0.07	35.50 ± 0.08	35.38 ± 0.17
Heart Weight/TL (mg/mm)	26.52 ± 0.35	37.78 ± 0.74 ***	37.68 ± 1.03 ***
Lung Wet Weight/TL (mg/mm)	11.07 ± 0.21	12.63 ± 0.28 **	11.88 ± 0.29
Kidney Weight/TL (mg/mm)	27.71 ± 0.75	45.50 ± 1.49 ***	46.75 ± 1.38 ***
Serum and blood parameter			
Lactate	1.36 ± 0.10	2.67 ± 0.17 ***	2.55 ± 0.13 ***
Serum NT-proBNP (pg/mL)	89.88 ± 8.02	203 ± 25.11 **	108.9 ± 26.2 ^#^
Echocardiography			
LV mass (mg)	818.6 ± 38.32	1274 ± 50.95 ***	1110 ± 37.05 ***^#^
LVEF (%)	69.02 ± 1.40	67.87 ± 0.97	71.06 ± 0.96
LVFS (%)	24.19 ± 0.16	25.01 ± 0.36	23.95 ± 0.21 ^#^
LVSV (µL)	290.1 ± 12.41	424.4 ± 21.3 ***	393.6 ± 12.56 ***
LVEDV (µL)	419.4 ± 13.06	627.3 ± 34.38 ***	554.7 ± 18.48 **
E/é	17.59 ± 0.74	24.62 ± 0.25 ***	19.69 ± 1.04 ^###^
E/A	1.15 ± 0.02	1.14 ± 0.02	1.41 ± 0.13
LVAW; d (mm)	1.70 ± 0.03	2.12 ± 0.03 ***	2.17 ± 0.09 ***
LVPW; d (mm)	1.52 ± 0.04	2.01 ± 0.06 ***	2.08 ± 0.10 ***
LVID; d (mm)	6.79 ± 0.09	7.73 ± 0.19 **	7.97 ± 0.18 ***
Septum; d (mm)	1.67 ± 0.05	2.25 ± 0.04 ***	1.94 ± 0.05 **^###^
Invasive Hemodynamics			
LVEDP (mmHg)	5.18 ± 0.56	10.54 ± 1.81 *	6.55 ± 1.19 ^#^
LVESP (mmHg)	100.3 ± 4.88	158.4 ± 6.06 ***	154 ± 5.68 ***
MAP in asc. Aorta (mmHg)	80.13 ± 3.09	107.3 ± 3.50 ***	110.6 ± 3.30 ***
LVEDV(µL)	379 ± 14.48	516 ± 33.30 **	462.8 ± 29.79
LVESV (µL)	164.2 ± 13.22	239.4 ± 21.29 *	189.6 ± 15.07
SW (mmHg x µL)	24,480 ± 1654	47,800 ± 2887 ***	47,933 ± 2974 ***
PVA (mmHg x µL)	97,767 ± 14,838	105,725 ± 16,540	97,767 ± 14,838
dV/dt max (µL/s)	5474 ± 447.7	5049 ± 524.4	5388 ± 791.8
dV/dt min (µL/s)	−5228 ± 430.5	−5551 ± 505.1	−6486 ± 551.3
Tau (ms)	19 ± 0.66	17.8 ± 0.55	18.37 ± 0.58
Slope LV-Ees (mmHg/µL)	0.17 ± 0.03	0.33 ± 0.07	0.36 ± 0.09
LV-stiffness constant β_w_	0.29 ± 0.08	0.74 ± 0.13 *	0.43 ± 0.07 ^#^

LV: left ventricle; NT-proBNP: N-terminal brain natriuretic peptide; LVEF: left ventricular ejection fraction; LVFS: left ventricular fractional shortening; LVSV: left ventricular stroke volume; LVEDV: left ventricular end-diastolic volume; LVAW;d: end diastolic left ventricular anterior wall thickness; LVPW;d: end diastolic left ventricular posterior wall thickness; LVID;d: end diastolic left ventricular inner diameter; LVEDP: left ventricular end diastolic pressure; LVESP: left ventricular end systolic pressure; MAP: mean arterial pressure; LVEDV: left ventricular end diastolic volume; LVESV: left ventricular end systolic volume; SW: stroke work; PVA: pressure-volume area; slope LV-E_es_: left ventricular end-systolic elastance. * *p* < 0.05, ** *p* < 0.005, *** *p* < 0.001 vs. Control and ^#^
*p* < 0.05, ^##^
*p* < 0.005, ^###^
*p* < 0.001 vs. HFpEF.

**Table 2 cells-12-02561-t002:** Effect of leucine supplementation on specific enzyme activities in HFpEF animals.

Enzyme	Control (*n* = 10)	HFpEF (*n* = 12)	HFpEF_Leu (*n* = 12)
Citrate synthase (CS; mU/mg)	254.2 ± 4.28	252.7 ± 6.56	260.1 ± 8.76
Lactate dehydrogenase (LDH; mU/mg)	877.9 ± 23.81	792.5 ± 19.18 *	890.8 ± 27.97 ^#^
Pyruvate kinase (PK; mU/mg)	35.77 ± 0.95	32.26 ± 0.67 *	31.54 ± 1.20 *

* *p* < 0.05 vs. Control and ^#^
*p* < 0.05 vs. HFpEF.

## Data Availability

The data that support the findings of this study are available from the corresponding author upon reasonable request.

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
