# Peer review of "Leucine Supplementation Improves Diastolic Function in HFpEF by HDAC4 Inhibition"

_cells, 2023, doi:10.3390/cells12212561_

Round 1

Reviewer 1 Report

Comments and Suggestions for Authors

This submitted manuscript is of interest since HFpEF will remain a significant threat to our health care systems. Like the authors mention, currently only the EMPEROR has resulted in significant improvements due to pharmacological treatment for patients. I only have some minor comments/suggestions for the authors.

- please mention in your methods section how NTproBNP was measured

- in your table 1 you show the characteristics of your animals. Please explain in more detail in the methods section the echocardiographic parameters (in particular LVM) were determined.

- Your results show a large degree of heterogeneity with regards to some outcomes (e.g. Fig 4B, 4C and 4E as well as Fig 5B and D). Could the authors speculate in the discussion section why treatment with Leucine would induce a response with such a large degree of variation?

- Leucine (and other BCAA) for this matter are available as supplementations for humans for a while now. Could the authors please speculate in the discussion section on potential barriers and opportunities for leucine treatment in patients with HFpEF?

Comments on the Quality of English Language

- Please consider not having paragraphs with one sentence

- Please consider restructuring sentences with "it"

Reviewer 2 Report

Comments and Suggestions for Authors

Dear Editors,

The following study submitted to Cells investigates the effects of leucine supplementation on myocardial function in a rat model of heart failure with HFpEF. The authors demonstrate that leucine supplementation improves diastolic function and reduces remodeling processes, providing valuable insights into potential therapeutic avenues.

However, there are several concerns that arise.

Major issues

-        How does the 3% leucine supplementation relate to humans when taking into account the human equivalent dose? Please state this in the discussion.

-        How do you explain the decrease in LV mass and septum size in HFpEF-Leu whilst the heart weigh / TL remains unchanged? How do you relate this to the decreased albeit not significant end diastolic volume?

-        To better characterize cardiac fibrosis and hypertrophy, exhaustive histological analysis is mandatory (Sirius red or Masson’s trichrome, and WGA).

-        Please discuss the results on enzymes activities.

-        In HDAC4 immunofluorescence, there seems to be an increase of expression in HFpEF and a decrease with leucine, rather than redistribution within the cells.

-        To further strengthen the mechanistic insights, it is essential to conduct in vitro experiments on cultured ventricular cardiomyocytes and fibroblasts w/o leucine and PKA inhibitor to assess leucine’s effect on hypertrophy, fibroblast activation and HDAC4 pathway. This is especially important in light of a recent publication where the authors demonstrated that cardiac-specific activation of BCAA oxidation does not protect from HF, and the cardioprotective effects of BCAA reside in their action on vascular resistance.

Minor issues

-        What is the rationale of using female instead of male ZSF1 rats? This should be stated in the methods section. The same applies for the 3% leucine supplementation.

-        How do you explain the effect of leucine supplementation on body weight?

-        Some recent literature on BCAA and heart failure have to be added: PMID: 36626303, PMID: 36856044. 

Round 2

Reviewer 2 Report

Comments and Suggestions for Authors

Dear Editors,

All of the issues have been adequately addressed by the authors.

Regarding the disparate findings for the heart weight and TL on one side and the LV mass and septum size on the other, it might be connected to the right ventricular mass or atrial mass.

In any case, I have no further comments and recommend the paper's publication.

Kind Regards